# Time-Varying Word Clouds

Category: Research

## ABSTRACT

We visualize time-varying text information with physically based simulation. Word-clouds are a popular means of visualizing the frequency of different concepts in a text document, but there is little work using text that has a time component, for instance, news feeds, twitter, or abstracts of papers published in a given journal or conference by year. We use physically simulated words that grow and shrink with time as an interactive web based visualization tool. We choose to use an existing 2D simulation framework Matter.js to develop the interface, with carefully designed forces to ensure a robust animation. We perform an informal user study to understand the ability of users to understand information presented in this dynamic way.

**Index Terms:** word cloud—time-varying—real time—dynamic visualization

## 1 INTRODUCTION

Word clouds are a method of visually portraying text-based information by using size as a means of depicting frequently occurring words in a document. Current word clouds that are available are mostly static and make use of text where time is not a variable. However, it can be useful to view various text documents compared using time as a variable. For example, visualizing keywords in a sports news magazine for every month throughout the year reveals which sports gain popularity over the course of the year (e.g., February may have more football due to the Superbowl versus May and June having more hockey and basketball due to playoffs). We design a system that allows us to observe these changes using a dynamic time-varying word cloud, where the words grow and shrink in size depending on the point in time. This way of displaying information allows for a new and engaging method of interpreting and comparing how text-based information changes over time.

Previous work has explored methods for visualizing time-varying text data. For example, Cui et al. [3] created a time-based word cloud coupled with a chart that lets the user see different time points of the word cloud. However, this time-based word cloud is not continuous, but presents different static word clouds at discrete points in time. Therefore, the change is only evident at each point in time but not evident between any the time points. What we propose allows the user to see a gradual visualization of change over time because the words slowly grow and shrink depending on their frequency at any point in time.

In order to create a dynamic word cloud with this type of behaviour, we use a physics based simulation with a collection of carefully designed forces that has words grow, shrink, and physically interact with each other while staying organized in a cluster. Our secondary goal is to create a web-based interactive tool available to everyone. Thus, we use a JavaScript physics engine called Matter.js with words existing as rigid bodies in the simulation world, and where the user can interact with different words (dragging with the mouse), and either play time forward or change the time with a slider interface. This real-time and dynamic approach is our contribution in comparison to other existing word clouds techniques.

Figure 1 shows an example of the software being used to visualize word trends of the SIGGRAPH annual conferences from 1974 to 2017. Words are forced to the center of the canvas and the current year is shown at bottom right. As time moves forward or backwards, these words will grow and shrink respectively depending on their

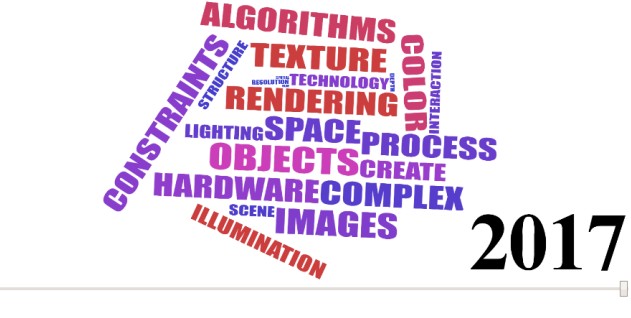

SIGGRAPH Yearly Conferences: Paper Titles and Abstracts

add    reset    play

Figure 1: A snapshot of our method visualizing titles and abstracts from SIGGRAPH technical papers.

frequency at that point in time. See accompanying demo video to view the software running this in real-time.

Finally, we try to understand the advantages and limitations of a continuous temporal word cloud through a user study. We attempt to answer the following question: does our dynamic word cloud allow for a quicker or better interpretation of the data compared to conventional techniques? We observe that our method does give users the ability to identify importance and variance while maintaining its aesthetic and visually stimulating method of portrayal. Although it is not part of our study, we believe that our approach could be effective for building an understanding of data that is changing rapidly over a short period of time. This kind of data would otherwise require many discrete and small time-steps to get a proper idea of the evolution, requiring numerous static word clouds if using the software of Cui et al. [3]. Using a continuous method allows for an easier interpretation by the viewer and a more efficient way of visualizing the data.

## 2 RELATED WORK

There has not been much research on the visualization of time-based information with word clouds. There are numerous static word clouds readily available to all online, but to the best of our knowledge, none have the ability to dynamically visualize data (time-varying or otherwise).

The most notable research that does consider time as a variable is that of Cui et al. [3], which specifically addresses the portrayal of temporal content evolution of a set of documents. In their work, the change in word cloud is visualized through the help of a trend chart. This chart has the purpose of indicating which one of several periods of time have more significant word clouds, where this significance is simply deduced by how much data is gathered from that time point. Using this method, the viewer is able to observe the different static word clouds at given point in time and compare one time point with another time point. To simplify this comparison, the same words are maintained in different clouds at the same location, regardless of a change in size. In addition, words of similar words definitions and occurrence over time are also grouped together and maintain the same location in the cloud, which allows for better interpretation. Finally, they also add color to the background of each

word to provide further information on the temporal quality of each word. Different colors signify if they are present in the previous cloud, the next one, both, or unique to this cloud in the timeline. In contrast, with our physics based simulation approach, the user can interact with words as they change size and interact with one another.

In follow up work, Cui et al. [2] further explored different methods in order to better understand evolving topics in text. They implemented what they termed TextFlow to demonstrate how a particular topic develops gradually and evolves in a set of time-dependent documents using a text-visualization method known as a river-flow-based visual metaphor. This visual representation allows for quick understanding of the topic flow and illustrates to the viewer at what point in time as well as why a splitting and merging of topics is triggered.

Wang et al. [13] address broadly the problem of visualizing time-varying data and suggest effective techniques to present and comprehend this type of data. As stated by Liu et al. [7] in their survey of information visualization, using word cloud representations and river metaphors have widely been used in attempts to improve the visualization of the evolution of topics in a set of documents. Another version of the river metaphor was implemented by Dubinko et al. [4] with the purpose of visualizing tags associated with photos posted in social media and their evolution over time.

Seyfert and Viola [12] create a dynamic word cloud where, for every point in time, each word gets a certain color and tilt in angle depending on if the word has grown or shrunk compared to the previous time point. Russel [9] graphs user tags over time on a web bookmarking service. Horizontal lines in the graph depict non-changing tags over time whereas diagonal lines represent changes in the tags. According to Russel, a change in the tags is attributed to either a change in content on the site, a change of description of the content or a change of behaviour by the users.

Chi et al. [1] created a word cloud with a morphable boundary over time to ease interpretation. For instance, the growth of a human over its life span, starting as a baby and growing into an adult. The word cloud is within the boundary of the shape of a baby and over time that boundary becomes the shape of a grown man while the word cloud inside also changes depending on the keywords associated with that phase of a human's life.

Kane [5] uses a physics engine in Python called Box2D to create a live depiction of how the words grow and shrink over time. This is a very similar project to ours, with the exception that it is not on the web, not interactive, and the words do not stack up onto each other, they simply hover in the air.

A problem that is also related to creating word clouds is the problem of packing shapes. Saputra et al. [10] use repulsion forces to fill up shapes with a set of given elements in a project called RepulsionPak. Each element in the target container shape considered as a mass-spring system, with the it's body being mapped by a triangle mesh. Initial random placement of these elements in small size is done and then, by iteration, the white space between them gets eliminated. This is done by deformation and growth of these elements with the help of repulsion forces, which balance both the level of deformation of each element and the tightness of the packing. The mesh map allows for accurate packing and prevents overlapping and leaving the container boundary. Saptura et al. [11] also did a similar packing project using flow to pack container shapes. FlowPak uses vector fields to shape, deform and stretch elements from a small set of shapes. Like in RepulsionPak, a series of iterations allows for gradual and accurate packing until the desired container shape is filled.

## 3 METHOD

The project was put together with the help of two JavaScript libraries: Matter.js and p5.js. The first is a physics engine, which is the backbone of the project. Each word is in fact a body in the engine's world. Bodies in this world can interact with each other, can be subject to forces like gravity, and can be given constraints. The latter is a processing library, which allows for customized rendering and ease of HTML functions like adding sliders and buttons onto the web page. It was necessary to allow for personal changes to the canvas and the rendering. Matter.js has its own renderer, however it is not very flexible and cannot complete many of the desired functions required for this project. For example, it does not have the capability to add word textures onto the bodies, and does not have the ability of writing the date on the canvas.

### 3.1 Design

There are many design decisions we, as creators, had to make to create an effective dynamic word cloud. First of all, we chose the layout of the words to be centered in the screen, and hover, like a cloud. Initially, we made words succumb to gravity and have a static ground and static walls to keep them in the frame as seen in Figure 3. However, we moved away from the use of gravity to allow for a more conventional "word cloud look". Ultimately we let larger words (more frequent words) have a greater attraction to the center as opposed to smaller words, therefore the more important words tend to the middle and the less important ones to the outside (see Figure 1).

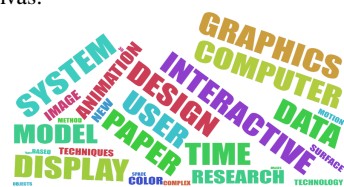

Figure 3: Gravity determines the final arrangement of words in this example, and words have random colors.

Another design decision was made with respect to the colors. We decided to create a variance heat map for the words, where the variance over the whole time span for each word is calculated, as opposed to simply random colors like in Figure 3. The color red is attributed to low variance words and blue to high variance words. If a word is somewhere in the middle, it is given a shade of purple, where the shade is scaled depending on its variance. For example, if it is more on the high variance side, it will be a more reddish purple.

Furthermore, during the creation of the word cloud, we noticed that words tend to flip upside down due to contact with other bodies. This is not ideal, as flipped words are difficult to read. To solve this problem, we created an inverse rotation force that is activated in the condition where the angle of the body is larger or smaller than $\pi/2$ and $-\pi/2$ radians respectively.

Finally, we decided to add interactive tools, such as mouse control over the words, if one wishes to move around the words with their mouse. In addition, a slider is presented below the graph to allow the user to slide through different time points quickly and easily. When using the slider, one can easily observe how words grow and shrink over time. There are also buttons are added for increased interaction. An "add" button allows to add more words to the screen, a "reset" button allows the user to clear the canvas and re-add words, and a "play" button lets the user sit back and watch the time period increment and observe the trends over time. Also, the p5 function "MouseWheel" is used to enable functions using the mouse wheel. When lots of words are added to the canvas, some words are outside the frame. Therefore, using the mouse wheel, the user can scroll and zoom in and out. This allows to capture all the words in the canvas at a larger scale.

We also consulted Munzner's book [8] to help validate certain decisions. She describes techniques and methods to consider when implementing computer-based visualization systems. For example, a change in a visualization system is important when considering large and more complex data to avoid viewing the information in an overwhelming manner. In other words, having a changing tool to demonstrate all the data as opposed to featuring all the data in one instance eases interpretation of the information. This allows us to

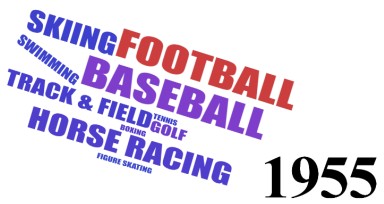 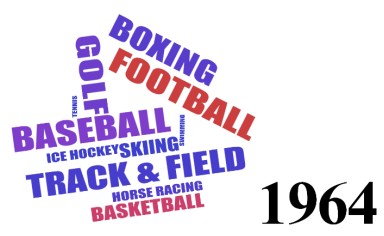 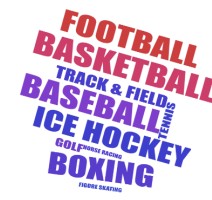

Figure 2: Word cloud evolution of the sports featured on Sports Illustrated at three time points

validate our idea of having a time-varying cloud. Another example is when she states that zooming is very useful when considering large datasets, to allow the user to change views between smaller subsets and the big picture. She also says that validation is very difficult when it comes to visualization, as there are so many questions to be asked regarding the system. Does it work? Is it really better than other methods? What does better even mean? Thus, despite not having much information regarding the effectiveness of our method, we can at least justify certain decisions.

## 3.2 Word Placement

Words are added to the world by the click of an "add" button on the screen. The purpose of this is to allow the user to decide how many words are desired to be on the screen since too many words can cause the data to be unclear and hard to interpret.

Once the button is clicked, a word is created and added to the world inside the canvas, as stated before. This is purely aesthetic and was simply a choice we made to give a stimulating effect.

Due to the Hookean force added to each body, the farther they are from the center, the larger the force towards the center. The horizontal coordinate of its placement is simply randomized over the span of the canvas' width.

## 3.3 Size of Words

Initially, the size of the word is a simple proportional function, where the initial frequency is scaled compared to the maximum possible frequency, and the font size is the same proportion to the max font size, that is,

$$F_i = \frac{f_i F_{max}}{f_{max}}. \tag{1}$$

Here, $F$ represents the font size, and $f$ represents frequency. The initial and max frequencies are retrieved from the text files containing the data, and the max font size is a parameter stated at the beginning of the program.

Once the words are added to the canvas, they must be scaled to the correct size at different time points. The target scaling factor $T$ is computed at every time step based on the desired height and the change in height depending on the current height,

$$h_{t+1} = \frac{f_{t+1} f_{max}}{k},$$
$$h_{\Delta t} = \alpha h_t + (1 - \alpha) h_{t+1},$$
$$T = \frac{h_{\Delta t}}{h_t}.$$

Here, $h$ represents the height at certain time points, where $t$ is for the current time point, $t + 1$ is the next time point, and $\Delta t$ is for the change between the two. In addition, the current, the next, and the max frequencies are required, and $k$ is a scaling constant. The parameter $\alpha$ controls the growth and shrink rate. This factor is added to allow for smoother size changes, as the words will grow and shrink at an inverse exponential rate.

## 4 EXAMPLES AND DISCUSSION

The main goal of this project was to create an interactive, in-browser visualization tool. The following results demonstrate a working dynamic word cloud that allows for visual interpretation of information.

The software was run on four separate files, each of different sizes and and time periods. The first one is a representation of the word trends of the SIGGRAPH yearly conferences from 1974 to 2017 (example used in Figure 1). The second is a count of all the instances of a certain sport featured on the cover of the weekly issue of the Sports Illustrated magazine, from 1955 to 2013. The third is all of the keywords from every month's edition of the American Association for the Advancement of Science (AAAS) magazine from January 1980 to August 2014. The last is the number of COVID-19 cases per country per day since the beginning of 2020. Each set of documents has a unique aspect which allows a different analysis. For instance, the sports data contains much fewer words compared to the other three data sets, however, it spans a larger time period, which is very interesting. On the other hand, the AAAS magazine data is taken per month not per year, therefore despite the time period being shorter, the slider spans many more points in time.

## 4.1 Examples

Figure 2 demonstrate an example of the software using the sports data. In the figure, we see the progression of three different time points and can see the change of trends depending on the year quite easily. In addition, looking at the slider we can see an indication of the current date on the timeline for the word cloud, as well as the actual date written on the bottom right. The changes between each frame happen organically; words slowly grow or shrink and get pushed around to make room. This example works nicely, as the information is quite easy to understand. For instance, in 1955, horse racing and skiing are quite large, therefore had many more appearances. In later years however, they are smaller or even disappear, which indicates that they were never even featured those years. Following just one topic, for instance, ice hockey is not even visible in 1955, but gradually becomes larger due to its increase in popularity over the second half of the century. In 1967, there was an expansion which doubled the number of teams in the National Hockey League, which explains why the tag is present in the second figure. In addition, in the third snapshot, representing the year 1980, it has become one the most prominent sports. This makes sense as this is the year "Miracle on Ice" took place, an iconic Olympic championship game still mentioned to this day. This example of the word cloud shows that this simple and aesthetic method of viewing information works. In a live presentation, starting from the first cloud in 1955 and simply clicking "play" would allow the time to slowly increment each year and the progression over 59 years would be observable in a short period of time.

Figure 4 demonstrates another example frame of the software mid-use using the AAAS data. This example tracks the evolution of many more words than the one above, therefore the contrast between this and the previous can be observed. Having more tags produces more immediate information, however it is more difficult

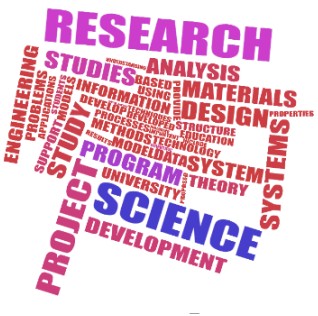

Figure 4: Visualization of word frequency changing monthly in American Association for the Advancement of Science (AAAS) magazine keywords.

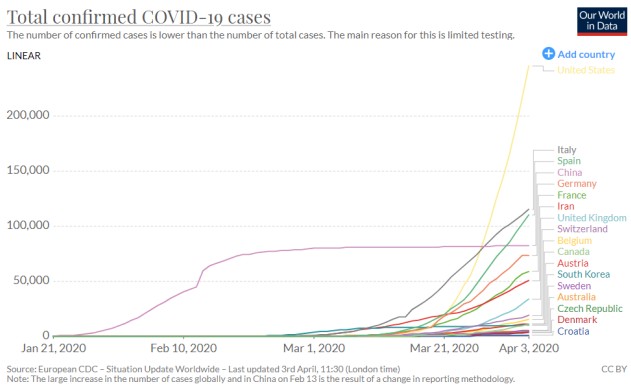

Figure 5: Snapshot of interactive tool displayed in the "Coronavirus Disease" article on the Our World in Data website [6].

to discern the details due to the large number of words. In contrast, when presented with less tags, it allows for easier, quicker, and more accurate processing of the details, but contains less overall information.

Finally, another example of the software can be observed in Figure 7, representing the number of COVID-19 cases per country per day since the beginning of the pandemic. This example is different than the others, as it represents a demographic change over time as opposed to the other three data sets which represent news sources and publications. Again, three different time points were taken to demonstrate the trends over the years. This example is a perfect demonstration of how simpler and more efficient this method of visualizing information is than other methods usually used for this kind of data. In the article from the Our World in Data website [6], where the data for this example was taken from, the authors discuss how countries have been evolving in numbers and they used a line graph as a visual representations to aid the readers in in viewing the data. Figure 5 is a snapshot of their interactive tool to demonstrate the number of cases over the given time period.

This graph represents the exact same countries as the word cloud, yet it is quite difficult to analyze the data in the graph. The lines are very intertwined and overlapping, and it becomes visually hard to follow. Whereas, using a dynamic word cloud allows the user to easily view which countries get larger over time. This type of visualization can also be more appropriate when presented on a screen to a group such as at conferences or workshops in which a close examination of a complex graph is not always possible. In addition, clearly all countries have a high variance in number of cases throughout the time period, therefore a different configuration for the colors was implemented. In this example, we set the color based on the date when they first passed 10 deaths. We used red to represent the first countries to surpass 10 deaths, blue to represent the last ones, and again, shades of purple to represent the ones in between. This enables the cloud to display more important information pertaining to the pandemic that is easy to interpret by the user.

### 4.2 Discussion

This project aimed at creating an information visualization tool to enable users to comprehend time-dependent data and view how it changes over time. The above examples demonstrate the capabilities of this software and how the information is displayed through the means of a real-time dynamic word cloud. Apart from representing changes over time by analyzing word frequency or trends in documents, this application can be useful for the representation of demographic data as well. In particular, a dynamic cloud is an interesting and clear way to represent demographic data that explores large geographical areas (such as many cities or countries). Economy related data such as changes in income for various populations,

or changes in house prices in various cities would also be well represented this way. While graphs tend to be more accurate and show specific numbers they are less effective in providing a visual representation when there are many factors in one graph representation, as seen in the example above. Another advantage to a real-time dynamic word cloud is that because it is visual and concrete it is easier to understand when data is presented to large groups or for non-scientific purposes, such as in the media. Using a dynamic word cloud can quickly demonstrate a particular trend in an interesting and stimulating way that can be easily interpreted.

### 5 IMPLEMENTATION

In this section, we demonstrate in detail how the project was implemented in JavaScript using libraries Matter.js and p5.js.

### 5.1 Matter.js

Every word added to the world is in reality a bounding rectangular box (see Figure 6). The dimensions of each word are calculated and depend on the initial frequency of the word when it is added to the world. Once added, the "scale" function from the Matter.js library is used to adjust their size depending on their changing frequencies over time. A mouse

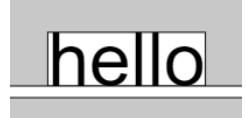

Figure 6: Bounding box enclosing the word

constraint is also created to allow mouse interactions with the words themselves. With a simple click and drag, the user can move the words to a desired location on the screen.

### 5.2 p5.js

The p5.js library is used to "draw" over the Matter.js world that is created. Without the Matter.js renderer, it is impossible to see the world that is created, therefore the work done by this library is simply to mirror everything that is done by the physics engine so that it becomes visible. The canvas was created in WebGL mode, which is a mode that allows 3D rendering and textures. The 3D was not necessary for our purposes, although textures are a crucial aspect of this project.

First, a canvas is created with the desired dimensions, the same dimensions as the Matter world. In addition, every time a body is added, p5 creates an identical shape, in this case a rectangle, that follows the coordinates of the Matter body. Alone, the shape cannot interact with anything as it is simply a drawing, but since it is "attached" to a physical body, it mimics all of its motions. When this identical rectangle is created, it also prints a texture onto it. Using a p5 function called "createGraphics", the desired word is made

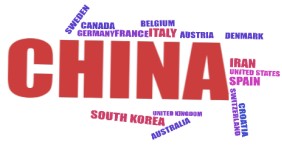
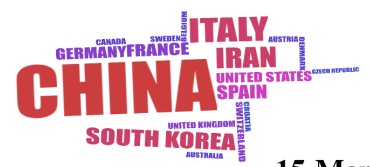
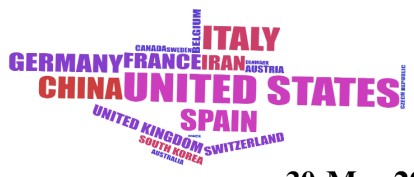

**27-Feb-20**  **15-Mar-20**  **30-Mar-20**

Figure 7: Three snapshots showing the evolution of a word-cloud visualizing the number of COVID-19 cases by country.

into a texture and then scaled to the desired size to be pasted onto the rectangle. The background of the texture is transparent, and the word is given a color depending on its variance over time, therefore all that is remaining in the p5 canvas is a word that falls from the sky and that can interact with other ones.

Finally, with the help of p5.js functions, interactive tools are created. A slider allowing to move through different time points is added along with three buttons: the "add", "reset", and "play" buttons. The current time point is also drawn on the bottom right of the canvas (see Figure 1).

### 5.3 Optimization

It is important to note that there are a few imperfections with the software at this point, related to the optimization of computation. A few factors can cause this program to not run smoothly and have been taken into account when creating this project. The computer it was run on was an ASUS laptop (3.00 GHz AMD A9-9420 RADEON R5 processor, 8.00 GB RAM).

Initially, words were meant to be soft bodies as opposed to rigid bodies like they are now. This difference was purely visual, to give the words a bouncing and twisting effect and to allow them to better fill in gaps between words, as rigid bodies sometimes create large empty

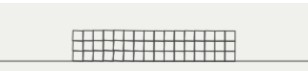

Figure 8: Composite of small squares to allow many degrees of freedom for the body of the word as seen from the Matter.js renderer

spaces. However, these bodies were actually composites of many small rigid squares (each a degree of freedom) connected by a constraint mesh. Therefore, resolving collisions required lots of computational power (see Figure 8).

The way the JavaScript physics engine works is that it takes every body in the world and checks if it collides with another body. This means that every square within one word had to get checked, so after adding only six or seven words, the program began to run very slowly. In addition, soft bodies tended to invert or overlap on each other, which posed another problem. Therefore, given our capabilities and time restrictions, a change was made and the skeleton of the words became one large rectangular bounding box, which is much easier to compute for the physics engine.

In addition, for the purposes of this project, the input file is simply a text file, not the actual set of documents. The set of documents are processed in Python and the results are copied into these text files. When the program is initialized, using the p5.js function "preload", the text files are loaded and parsed before the program even appears, and this is done very quickly, almost imperceptibly. Since the goal of the project is to create a dynamic word cloud and not a data parsing software, this was done to allow for more efficient run time.

### 6 USER STUDY

We carried out an informal user study to comprehend the efficiency of this method of information visualization. The goal was understand

to what level information is retained from this method of presentation. It is very interesting to see how efficient a dynamic word cloud is at demonstrating trends and getting the viewer to absorb information.

**Method** The user study was presented as follows: first, the users would be prompted to view our dynamic word cloud and given instructions for interactions and a time limit. The users would then be asked questions testing their knowledge on what they just learned by using the software. After the users completed this section, they moved on to the next part, which consisted of a very similar procedure, but using the more conventional multi-line graph instead (like in Figure 5).

**Observers** 10 subjects participated in the study. All subjects were undergraduate university students.

**Procedure** In the first section of the study, the data presented is the keywords from the titles and abstracts from the SIGGRAPH yearly conferences. Three multiple choice answer questions were asked:

1) Select the top 3 most popular words throughout the timeline.
2) Which of these words has the highest variance?
3) Which of these words has the lowest variance?

In the second section, a conventional graph was presented to the users. This graph shows how the norms of household technology changed over time by representing the percentage of households in the US that use a certain technology from 1975 to 2019. After the users analyzed the graph for a set amount of time, they were asked the same three questions.

### 6.1 Results

Tables 1 and 2 demonstrate the results from the user study. Table 1 represents the question asking for the three most popular words and Table 2 represents the questions regarding the highest and lowest varying words throughout the timeline.

Regarding the dynamic word cloud section, in selecting the three most frequently occurring words, the top two were demonstrated an 80% success rate, while the third showed a lower 60% success rate. For question three regarding the lowest varying word, 70% of the participants answered accurately. However, when identifying the highest varying word, success rate was only at 30%. We believe this is due to the difficult nature of keeping track of things that are changing. It is important to note that questions one and three only asked the user about the largest words and the words that change the least respectively, therefore easier for the user to retain.

In the multi-line graph section of the user study, when asked about the three most popular words (question one), responses indicated an almost 100% accuracy in selecting the most frequent word, but for the next two words, the accuracy diminished substantially. This could be due to the fact that the numerous lines in the graph are very intertwined, and it is difficult to discern which ones stand out as

Table 1: Success rate in percentage for identifying top three most frequently occurring words

| Question | Section 1: Word Cloud Success Rate (%) | Section 2: Multi-Line Graph Success Rate (%) |
|---|---|---|
| Identifying Most Popular Word | 80% | 90% |
| Identifying Second Most Popular Word | 80% | 60% |
| Identifying Third Most Popular Word | 60% | 50% |

Table 2: Success rate in percentage for identifying highest and lowest varying words

| Question | Section 1: Word Cloud Success Rate (%) | Section 2: Multi-Line Graph Success Rate (%) |
|---|---|---|
| Identifying Highest Variance | 30% | 70% |
| Identifying Lowest Variance | 22.2% | 66.7% |

the most popular. For the highest and lowest variance selections, a similar pattern as in section one was observed. Responses indicated a low accuracy level for selecting the highest varying word and a relatively high accuracy for selecting the lowest varying word. Like above, we believe this is due to the difficulty of absorbing complex data versus simple data. The highest varying word is depicted by the most vertical line in the graph, thus crossing many other lines, whereas the least varying word is depicted by the most horizontal line, having less intersecting lines. Therefore, this line is more visible and easier to retain.

From the data collected in this informal user study, we can conclude that the dynamic word cloud can have some advantages over a conventional multi-line graph, but only pertaining to a specific set of queries. Clearly, if asking for a certain value at a given point in time, then the graph is much more pertinent. However, based on this small sample study, the dynamic word cloud enabled the user to grasp the popular trends fairly well and may even have a slight advantage for detecting the most constant words.

With respect to limitations of this study, we recognize the relatively small sample size to gather the data. In addition, one of the purposes for creating this cloud was to create a more visually stimulating and engaging method of portraying information. While we did not ask participants to rate these qualities when analyze the data, we do believe that the interactive nature of the software and the changing visuals tend to be more pleasing to viewers. This may be something to be explored further.

## 7 CONCLUSION

In conclusion, we present a method for portraying time-varying text information in a dynamic word cloud. This allows for the user to view trends from a set of time-dependent documents in a unique manner. While the method may not be the most optimal manner of displaying data, it gives a potentially more entertaining and stimulating visual effect than other methods of visualizing data (e.g., graphs). We believe that a dynamic word cloud is ideal for a live presentation where the viewers are kept engaged and interested by this active data analysis tool or for a user to interact with on his or her computer.

### 7.1 Future Work

Initially, the goal was to create an online interactive tool available to all. Therefore, what would be left to do is to create a web space where a data parsing software is available to take as an input a set of time-dependent documents that parse the n-grams into the correct text file format. At the moment, the data needs to be in a specific format within a text file and all the time data is required from the beginning of the program startup. A continuous input would be a great improvement, as it would be able to adjust given real-time data. Or, more simply, create a parsing software that allows the user to input a set of documents (news articles, magazines, etc) and have the program read those and then create the word cloud without the need for the text file and its format. From there, anyone would be able

to input their own set of documents each with a timestamp and the software will be able to read and portray the information received through a dynamic word cloud.

In addition, fixing the run time issues of collision checking would greatly enhance the project. Using broad phase collision detection could be a solution to this, however Matter.js must be modified in a manner to make this possible, which can be a whole project in itself. From there, soft body words can even be implemented to complete the initial desired image. Fixing the physical errors of inverted and overlapping words would also be required.

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
