# OpenReview forum: "Time-Varying Word Clouds"
_graphicsinterface.org/Graphics_Interface/2020/Conference — Submitted to GI 2020_

### Official Review · AnonReviewer2 · 2020-04-20
**So fun!**

**Rating:** 8
**Confidence:** 2

**Review:**

When I saw the video for this paper, my reaction was "this is so fun! I wish I had done this!"

I feel like the *concept* is in itself an extremely strong part of this submission. Independent of implementation details, I feel like the concept itself is an exciting point of merit for publication of this submission.

It's interesting that when the paper discusses prior work, a lot of that prior work is clearly targeting printed media (or other non-dynamic media). The reason that this paper works is because the medium is dynamic. If I were limited to a hardcopy printed paper, I couldn't use this method. So in some sense, the novelty here is "what can we do in an interactive setting." It's not a "we can do better than the previous work" rather a "we consider a different problem than the prior work."

The actual implementation is maybe a bit more "permissive" than it needs to be. In principle, to visualize a time varying word cloud I need a temporally coherent display of the words as they are born, change in prominence, and die. The physical simulation is a very good metaphor for getting that coherence and getting all the words to fit together. On the other hand, it goes too far, in a sense, since the momentum, wiggling, and ability to shuffle the words around are all distractions from studying the temporal trend. Were these latter features just an artifact of using a physical simulation? How could we keep the benefits of the simulation while factoring out these distractions?

---

### Official Review · AnonReviewer1 · 2020-04-20
**nice problem, not very developed solution**

**Rating:** 4
**Confidence:** 4

**Review:**

This paper seeks a temporally evolving word-cloud visualization. Given frequency data for a set of words, the method computes target sizes and has the words grow and shrink. It uses physical simulation (collisions and Hookean springs) to manipulate word placement. A brief informal user study is inconclusive about the usefulness of time-varying word clouds.

The paper identifies an excellent problem and makes a nice first attempt to address it. I am not convinced by the technique and the user study adds little value. Overall, I would encourage the authors to continue work in this direction but I think the work is not quite mature enough for publication. The design presented does not quite crack the problem: the physical forces are a nice ingredient but as presented are not sufficient to deal with the layout, packing, and alignment issues that the problem conjures. The user study does not show the visualization to be superior to a conventional line graph, and the data mostly points in the opposite direction. How do the authors envision the visualization being used? The questions asked for Table 2 do not seem like ones that users would really be interested in.

The discussion is a bit too tied to the specific libraries and web technologies used. These are not details of great interest. They should be disclosed, yes, but it would be better to concentrate on the design aspects and algorithmic aspects independent of implementation.

The interactive aspects of the method are not much justified or explored, leaving this reader wondering why they were discussed at all. The core problem seemed to be to generate a temporal visualization of a dataset. Adding interaction might make a more useful tool for data exploration, but seems external to the initial algorithmic problem. Since the interactions described have little algorithmic interest and were not evaluated, they can safely be omitted from the discussion. This is not to say that the researchers should give up on interactions, only that there should be a clear purpose and contribution if they are in the paper.

Specific comments:

"Kane [5] uses a physics engine in Python called Box2D to create a live depiction of how the words grow and shrink over time. This is a very similar project to ours, with the exception that it is not on the web, not interactive, and the words do not stack up onto each other, they simply hover in the air" : Some of the criticisms here seem unfair. "Not on the web" is an extremely weak point, not addressing the method at all. "[the words] simply hover": this is a design decision, and it is not clear (no case is made) that stacking is better.

"dynamic word cloud ... may even have a slight advantage for  detecting the most constant words." Why do you think so? Table 2  reports a 22% success rate for the word cloud, vs. 67% for the  graph. Is there an error in how the data is reported, or is my  interpretation of the data mistaken somehow? In my reading, this is a clear win for the graph.

Section 5.3 is largely about a failed design direction and can be cut. The lack of broad-phase collision detection in Matter.js is not broadly interesting and the researchers should not feel restricted to using Matter (many other physics engines exist).

---

### Official Review · AnonReviewer3 · 2020-04-20
**great introduction but a design and study section that lacks arguments and descriptions**

**Rating:** 3
**Confidence:** 4

**Review:**

This paper presents time-varying word-clouds that uses physically based simulation. The words in the word cloud change their size based on the data at a given time point. By using a time slider users can see the evolution of the frequency of words over time. The color of each element is selected based on the variance of word frequency over the time period of the dataset. The authors allow the user to interactively select and move words around.

I really liked the introduction and how it introduced the problem. The introduction made me want to continue reading. However, the rest of the paper did not follow on that promise.

The design decisions are unfortunately not well documented and the reason for certain decisions are vague. For example the first sentence about the design says "... effective dynamic word cloud", but there are no requirements of what effective should be or what an effective dynamic word cloud could be. I'm intrigued by the authors decision of making their tool interactive to move words around. There is no clear argument and reasoning of why this could be helpful. What could this be helpful for? Are maybe any task that would benefit from such a functionality?

The comparison between interactive word clouds and line graphs in the discussion is vague and not substantiated. Why are line graphs not visual and concrete?

The section on implementation details could be made much more succinct. The issues mentioned are not very relevant and interesting. Especially the section on optimization hangs in the air and does not connect to any other parts of the paper. Why is this relevant to the reader? I would suggest to use any gained space to describe and analyze the "user study" in more details. Because as it s now the study is not giving much arguments for using time varying word clouds.

The informal "user study" is described very minimalistically and the results suggest that line graphs are better suited for the described tasks. If I would just look at the two tables I would choose the line graph over the dynamic word cloud. The line graph shows all the data at once where as a dynamic word cloud needs manual interaction and a user only gets a moment view of a specific time point. Using the slider can probably show which words stay and that some words are appearing and disappearing if the set of words stays constant but this is not always the case. A dynamic word cloud might be useful for some settings of data but which one?

Finally, the conclusion and future work could be directed more towards what are the questions that are still unanswered. What about improving and making a formal and correct study? Current suggested future work is engineering but I am missing interesting research question that might have come up during this project.

---

### Meta-Review · Area_Chair1 · 2020-04-23

**Recommendation:** Reject
**Confidence:** 5

**Metareview:**

R1 and R3 are for rejecting the paper, R2 finds it a clear accept. However, R2 is very uncertain with his review. All the reviewer think the problem is interesting and that it is a problem worth looking at. However, R1 and R3 have concerns with the study, find the motivation of the technique lacking, and miss a clear elaboration of the design decision employed to get to this technique. In the study time-varying word clouds are compared to line graphs. There is no evidence for the technique being superior to line graphs even though the authors suggest otherwise. Furthermore, R3 hints at possible distractions from the physically based simulations but no discussion in the paper takes up this point. In summary the cons are major and I therefore recommend to reject this paper.

---

### Decision · Program_Chairs · 2020-04-25

Reject